# Effect of Higher Ethylene Levels Emitted by Shade-Avoider Plants on Neighboring Seedlings

**DOI:** 10.3390/plants13223212

**Published:** 2024-11-15

**Authors:** Mikel Urdin-Bravo, Angela Sanchez-Garcia, Manuel Rodriguez-Concepcion, Jaume F. Martinez-Garcia

**Affiliations:** Institute for Plant Molecular and Cell Biology (IBMCP), CSIC-Universitat Politècnica de València, 46022 Valencia, Spain; mikel.urdin@ibmcp.upv.es (M.U.-B.); angelasg@ibmcp.upv.es (A.S.-G.)

**Keywords:** *Arabidopsis thaliana*, *Cardamine hirsuta*, ethylene, proximity shade, tomato, volatiles

## Abstract

Plants of several species, including crops, change their volatilome when exposed to a low ratio of red to far-red light (low R/FR) that informs about the presence of nearby plants (i.e., proximity shade). In particular, the volatile hormone ethylene was shown to be produced at higher levels in response to the low R/FR signal in shade-avoider plants. Here, we show that the shade-tolerant species *Cardamine hirsuta* produces more ethylene than shade avoiders such as *Arabidopsis thaliana* (a close relative of *C. hirsuta*) and tomato (*Solanum lycopersicum*) under white light (W). However, exposure to low R/FR (specifically to FR-supplemented W, referred to as W+FR or simulated shade) resulted in only a slight increase in ethylene emission in *C. hirsuta* compared to shade avoiders. Stimulation of ethylene production by growing plants in media supplemented with 1-aminocyclopropane-1-carboxylate (ACC) resulted in reduced hypocotyl growth under W+FR in both *A. thaliana* and *C. hirsuta*. ACC-dependent ethylene production also repressed hypocotyl elongation under low W and in the dark in *C. hirsuta*. By contrast, in *A. thaliana*, ACC supplementation inhibited hypocotyl elongation in the dark but stimulated it under W. Most interestingly, elongation of dark-grown *A. thaliana* seedlings was also repressed by exposure to the volatiles released by ACC-grown *A. thaliana* or tomato plants. This observation suggests that increased ethylene levels in the headspace can indeed impact the development of nearby plants. Although the amount of ethylene released by ACC-grown plants to their headspace was much higher than that released by exposure to low R/FR, our results support a contribution of this volatile hormone on the communication of proximity shade conditions to neighboring plants.

## 1. Introduction

Plants typically grow in groups, forming communities of individuals from the same or different species. In crowded environments, plants may compete with their neighbors for essential resources, such as light, water, or nutrients. Plants can also benefit from nearby vegetation through mutualism, where both participants help each other, or facilitation, where at least one participant benefits without harming the other [1,2,3]. The mechanisms behind plant-to-plant communication supporting both negative or positive interactions involve generating, sensing, and transducing signals. These signals can be physical, such as light, or chemical, including exudates and volatile organic compounds (VOCs). Regardless of its nature, when a plant produces a signal that contains information for another one, we can refer to them as emitter and receiver plants, respectively. While some signals produced by emitter plants (e-plants) in response to environmental challenges are well known, our understanding of how these signals are integrated to elicit a response in receiver plants (r-plants) remains limited.

The proximity of vegetation can impact both light quantity and quality. Under a vegetation canopy, the overtopping green leaves strongly absorb red light (R) but reflect and transmit far-red light (FR). As a consequence, the light reaching plants growing in forest understories is less intense but also shows a reduced R-to-FR ratio (R/FR) compared to those growing in open spaces. In dense plant communities, FR reflected by neighboring plants also decreases R/FR but typically without changing light intensity. We refer to the first situation as canopy shade (involving lower light intensity and very low R/FR) and the second as proximity shade (same light intensity but low R/FR) [4,5]. Plants have evolved two main strategies to respond to the light signals informing about canopy and proximity shade originated from e-plants: avoidance and tolerance. Shade-avoider species, like the model plant *Arabidopsis thaliana*, display a set of responses collectively known as the shade-avoidance syndrome (SAS), including the promotion of elongation of organs to outgrow the neighbors, reduction of the levels of photosynthetic pigments to cope with actual and anticipated light shortages, and acceleration of flowering to ensure species survival when exposed to shade [4,5,6]. By contrast, shade-tolerant species, like *Geranium robertianum* and the *A. thaliana* relatives *Cardamine hirsuta*, *Arabis alpina, Nasturtium officinale*, and *Sisymbrium irio*, are slightly or not responsive to shade signals in terms of elongation or photosynthetic pigment accumulation [6,7,8].

The R/FR changes associated with the proximity of vegetation are perceived by the phytochrome photoreceptors, whose activity is regulated by the R/FR conditions. Among the five phytochromes identified in *A. thaliana*, phytochrome A (phyA) and B (phyB) are the most important in regulating the SAS responses. PhyA plays a negative role in elongation under very low R/FR conditions [9,10], while phyB inhibits elongation under high R/FR (when there is no shade) by inactivating PHYTOCHROME INTERACTING FACTORs (PIFs). PIFs are members of the basic helix–loop–helix transcription factor family that promote elongation growth [4,5]. After exposure to low R/FR, phyB is inactivated, and, consequently, PIF activity increases. Then, PIFs rapidly change the expression of hundreds of genes, leading to the hypocotyl elongation, including some regulating hormone levels [11,12,13,14]. Comparative analyses of *A. thaliana* and *C. hirsuta* showed that the molecular pathways sensing and transducing the low R/FR signals are quite conserved between shade-avoider and -tolerant plants. The attenuated responses to shade observed in the shade-tolerant *C. hirsuta* can be explained by, at least, an enhanced activity of negative regulators of the SAS, such as phyA and the PIF repressor HFR1 [6,15,16].

Plants release VOCs from their leaves, flowers, and fruits into the atmosphere, as well as from roots into the soil. These VOCs, mainly composed of secondary metabolites, such as terpenoids (also known as isoprenoids), fatty acid derivatives, benzenoids, and phenylpropanoids, serve a wide variety of functions [17]. VOC emissions may vary significantly among plant species to the extent that they can be used to classify species [18]. Some VOCs are synthesized constitutively, but others change in response to environmental signals. Exposure to low R/FR signals has been shown to alter the composition of the VOC blend emitted by *A. thaliana* and tomato (*Solanum lycopersicum*) plants [19,20]. These VOCs may potentially elicit responses in neighboring plants; hence, they might be involved in neighbor detection in crowded environments [21]. In such cases, the same plant initially functioning as an r-plant of a low R/FR signal can subsequently become an e-plant and emit VOC signals that may modulate the performance of neighboring r-plants.

One of the VOC components that change in response to low R/FR is ethylene, a volatile plant hormone whose levels rise in dense plant communities [21,22]. An analysis of a few shade-avoider species showed that the levels of ethylene increase under low R/FR conditions, but whether this response is also present in shade-tolerant species has not been explored yet. Importantly, the amount of ethylene accumulating within the atmosphere of dense stands of greenhouse-grown plants, like tobacco (*Nicotiana tabacum*) [22], is proposed to be high enough to trigger SAS responses, such as elongation [23]. Therefore, ethylene produced by shaded e-plants can contain information for neighboring r-plants, making it a likely signal in plant-to-plant communication in crowded stands. However, it is not demonstrated whether the ethylene emitted by one plant (instead of exogenously supplied) can affect nearby vegetation. In this work, we aimed (1) to determine how widely shade induces ethylene production by analyzing its production across shade-avoider and shade-tolerant species and (2) to investigate whether the ethylene emitted by one plant can be perceived and trigger a response in neighboring plants. Our results provide novel experimental evidence supporting a role for ethylene in plant-to-plant communication among nearby plants.

## 2. Results

### 2.1. Ethylene Production upon Exposure to Simulated Shade Is Attenuated in Shade-Tolerant Plants

Ethylene production has been shown to be activated by exposure to low R/FR in several shade-avoider species, but no information is available on shade-tolerant plants. To explore potential mechanisms, we measured ethylene levels in leaves from both shade-avoider or shade-tolerant plant species after exposure to FR-enriched white light (W + FR, herein also referred to as simulated shade). As shade avoiders, we used *Arabidopsis thaliana* Col-0 and tomato (*Solanum lycopersicum*) M82 and MicroTom accessions, whereas *Cardamine hirsuta* Ox was used as the shade-tolerant plant. Leaves from plants grown on soil under white light (W) were detached from the plants and immediately placed in airtight transparent glass tubes for illumination with either W or W+FR. The ethylene content in the headspace of the tubes containing the leaves was measured after 24 h. In all three shade-avoider accessions, leaves exposed to W+FR released much more ethylene in 24 h than their corresponding W-illuminated controls (Figure 1). By contrast, *C. hirsuta* leaves produced much more ethylene under W, and it hardly increased under W+FR (Figure 1). These results suggest that shade-induced ethylene production is reduced in shade-tolerant leaves compared to shade-avoider ones, perhaps because shade-tolerant plants already produce high amounts of ethylene under non-shaded conditions.

### 2.2. Endogenous Ethylene Impacts Hypocotyl Elongation in a Light-Dependent Manner

To investigate whether ethylene levels in shade-avoider and shade-tolerant plants might influence their shade response, we measured hypocotyl elongation in *A. thaliana* and *C. hirsuta* plants germinated and grown in media supplemented with 1-aminocyclopropane-1-carboxylate (ACC), an ethylene biosynthetic intermediate that promotes hormone production [24]. The *A. thaliana* ethylene-insensitive *ein2-5* mutant was used as a control, expecting no effect of ACC supplementation [24]. Seeds of the three genotypes were germinated on plates either supplemented or not with 10 μM ACC. After stratification, plates were incubated for 2 days under W and then either transferred to W+FR or left under W for 5 more days. In the absence of ACC, *A. thaliana* WT and *ein2-5* seedlings treated with W+FR showed longer hypocotyls than those grown under W, whereas hypocotyl length was similar in *C. hirsuta* seedlings exposed to either W or W+FR (Figure 2). Also, as expected, stimulation of ethylene production in ACC-grown seedlings had no effect on hypocotyl elongation of *ein2-5* seedlings grown under W or W+FR. In the case of *A. thaliana* WT seedlings, ACC increased hypocotyl length under W and reduced it under W+FR (Figure 2). By contrast, elongation of *C. hirsuta* hypocotyls was repressed by ACC treatment under both W and W+FR (Figure 2).

We next tested whether the differential effect of ACC supplementation on hypocotyl elongation of shade-avoider (*A. thaliana*) and shade-tolerant (*C. hirsuta*) plants could also be observed in seedlings growing under other light conditions. Seeds were germinated on plates with or without ACC and incubated for 7 days either in the dark or under different intensities of W (Figure 3). In darkness or under low W irradiation, ACC supplementation caused shorter hypocotyls in both *A. thaliana* and *C. hirsuta*. At higher W intensities, the presence of ACC still led to shorter hypocotyls in *C. hirsuta* but it resulted in longer *A. thaliana* seedlings (Figure 3). At intensities of 100 μmol m^−2^·s^−1^ or higher, ACC had no effect on *C. hirsuta*, while it still promoted hypocotyl elongation in *A. thaliana* seedlings (Figure 3).

### 2.3. Elongation of Dark-Grown A. thaliana Seedlings Is Repressed by Ethylene Released by Neighboring Plants

The results reported to this point suggested that the higher levels of ethylene released by shade-avoider e-plants in response to simulated shade (Figure 1) might have an impact on the development of neighboring r-plants. To investigate this possibility, we used plates with two separate compartments, allowing us to grow plants in separate media while sharing the same airspace. The idea was to grow seedlings releasing ethylene in the e-plant compartment and test the effect on the elongation of neighbors growing in the r-plant section. In the e-plant compartment, we placed about 15 tomato MicroTom seeds or 100 *A. thaliana* seeds on media with or without 10 μM ACC. The other compartment was used to grow *A. thaliana* without ACC as r-plants. Besides the Col WT, in the r-plant compartment, we included the *ein2-5* mutant and a few seeds of the *EBSn:GUS* line, which expresses the *GUS* gene under the control of an ethylene-induced promoter (a gift from the Alonso–Stepanova Lab). After incubating the plates in the dark for 4 days, we used the *EBSn:GUS* seedlings for GUS staining and measured the hypocotyl length of etiolated WT and *ein2-5* seedlings from the r-plant section (Figure 4). *EBSn:GUS* seedlings growing in the presence of volatiles emitted by ACC-grown tomato or *A. thaliana* seedlings showed stronger blue staining (i.e., higher GUS activity) than those exposed to volatiles from seedlings growing without ACC (Figure 4A). This result confirmed that enhanced ethylene levels released by seedlings in the e-plant compartment could be perceived and signaled by seedlings in the r-plant section of the same plate. Measurement of hypocotyl length showed that elongation of etiolated *A. thaliana* WT (but not *ein2-5*) seedlings growing in the r-plant compartment next to the *EBSn:GUS* markers was repressed when the e-plant section contained ACC-grown tomato or *A. thaliana* seedlings (Figure 4B). We therefore concluded that enhanced ethylene produced by e-plants can indeed influence the growth of nearby r-plants.

The repression of etiolated seedling elongation when ethylene production is endogenously enhanced by ACC feeding (Figure 3) is much stronger than that caused by the airborne ethylene released by neighboring e-plants (Figure 4). This was not surprising, as we expected that only a fraction of the extra ethylene produced by ACC supplementation (or W+FR treatment) would be released to the airspace and reach nearby plants. To quantify how much ethylene was released by ACC-treated seedlings, we germinated *A. thaliana* (Col-0) seeds in 50 mL short wide-mouth sample glass bottles containing 20 mL of media either supplemented or not with 10 μM ACC. Following stratification, the bottles with the seeds were incubated for 7 days under W, and then the amount of ethylene emitted to the headspace in 24 h by the seedlings growing in the bottle was measured (Figure 5). In a parallel experiment, the bottles with the stratified seeds were incubated for 2 days under W and then left under W or exposed to W+FR for 5 additional days. As expected, only bottles containing seedlings emitted measurable levels of the hormone to the headspace (Figure 5). In the absence of ACC supplementation, seedlings exposed to W+FR released about twice the amount of ethylene than W controls (Figure 5), similar to that previously observed using detached leaves (Figure 1). ACC supplementation resulted in a much more dramatic ethylene emission, reaching levels that were about 10-fold higher than those released by controls growing without ACC (Figure 5). Together, our data show that volatile ethylene emitted by e-plants exposed to low R/FR has the potential to alter the development of neighboring r-plants, even though the amounts are so low that it might not have a visible effect.

## 3. Discussion

By implementing SAS responses, shade-avoider species adjust their development when growing in high-plant-density or crowded environments. Although the shade-induced changes in development may vary with species and developmental stage, the most commonly analyzed SAS responses are the promotion of elongation, reduction in photosynthetic pigments, and induction of flowering. These responses optimize the growth, photosynthesis, and architecture of the shaded plant, enabling it to better compete with neighboring vegetation and counteract the potential light deprivation caused by nearby plants. This helps the plant to either escape from shaded areas, adapt to the new light conditions, or ensure reproduction for producing the next generation. Another type of shade-induced response is the increase in ethylene emission. In tobacco, ethylene production under low R/FR follows a diurnal pattern, with higher production rates observed during the light period, which subsides when R/FR levels increase again [22]. Other shade-avoidance species, including *A. thaliana*, *Brassica rapa*, and *Sorghum bicolor*, have been reported to stimulate ethylene production in response to low R/FR after different times of exposure to this light signal [25,26,27]. In our low-R/FR conditions, we also observed enhanced ethylene production not only in the shade-avoiders *A. thaliana* and tomato but also in the shade-tolerant *C. hirsuta*, although to a lower extent (Figure 1), suggesting that enhanced ethylene emission might be part of a core response shared by multiple species when exposed to low R/FR. However, the elevated ethylene levels emitted by *C. hirsuta* leaves even under non-shaded conditions, similar to those of *A. thaliana* under W+FR, suggest that enhanced ethylene emission might be an SAS-related response that is constitutively activated in this shade-tolerant species under high R/FR. This constitutively enhanced ethylene production might limit the additional ethylene emission after exposure to low R/FR in this shade-tolerant species, as observed in this work (Figure 1).

Studies using ethylene-insensitive lines have revealed a role for ethylene in shade-induced elongation in tobacco [22,28]. Our analyses of *A. thaliana* using ACC also support the involvement of ethylene and the signaling component EIN2 in the shade-induced hypocotyl elongation. Interestingly, ACC treatment had opposite effects on hypocotyl elongation in *A. thaliana* (Col-0) and in *C. hirsuta* (Ox) seedlings under W, whereas it inhibited this response in both species under W+FR (Figure 2). The effect of extra (ACC-dependent) endogenous ethylene on hypocotyl elongation might depend on the basal levels of the hormone present in shade-avoider and shade-tolerant plants. The higher basal levels found in *A. thaliana* under W+FR or in *C. hirsuta* under W or W+FR (Figure 1) might lead to hypocotyl length inhibition in the presence of ACC, i.e., when more ethylene is produced (Figure 2). However, the measurement of ethylene levels in the headspace of *A. thaliana* WT seedlings growing in airtight glass flasks (Figure 5) showed that this is probably not the case. Alternatively, ethylene responsiveness or sensibility might differ between *A. thaliana* and *C. hirsuta* (at least under W), and light conditions might have a role in these differences.

Previous reports indicate that hypocotyl elongation is promoted by ACC in light-grown seedlings [24,25,29,30] but inhibited in dark-grown seedlings [24,31]. Our work further demonstrates that (i) low R/FR reduces responsiveness to ethylene in light-grown *A. thaliana* seedlings (Figure 2), and (ii) there is a turning point where the effect of ethylene on *A. thaliana* hypocotyl elongation shifts from inhibitory (in darkness or under low W) to promoting (in higher W intensities) (Figure 3). Ethylene via ETHYLENE-INSENSITIVE 3 (EIN3) concomitantly activates two contrasting paths affecting hypocotyl growth [30]. One of them involves the transcription factor PIF3, which was shown to be required for ethylene-induced hypocotyl elongation in the light. The other involves the growth inhibiting ETHYLENE RESPONSE FACTOR 1 (ERF1). The balance between both was proposed to lead to ethylene-mediated growth inhibition in the dark and growth promotion in the light [30]. PIF activity was proposed to be reduced in *C. hirsuta* relative to *A. thaliana* [16], likely contributing to the different impact of ethylene-induced hypocotyl elongation in light in both species. It remains to be explored if EIN3 and/or ERF1 activities also differ between both species.

An aspect that has been underexplored is whether ethylene could function as a neighbor detection signal. In such a scenario, ethylene emitted by shaded e-plants would be expected to elicit a response in nearby r-plants. Ethylene produced by neighboring seedlings of tomato or Arabidopsis appears to be detected by dark-grown r-plants, as indicated by the GUS staining of the ethylene-specific *EBSn:GUS* marker line and the inhibition of hypocotyl elongation of *A. thaliana* WT seedlings. Although the effect of seedling-produced ethylene on this trait is mild, it seems specific, as no response was observed in the *ein2-5* mutant (Figure 4). In our experimental conditions, feeding seedlings with ACC results in a much higher ethylene production (about 10 times more) than non-ACC-treated seedlings exposed to W+FR (Figure 5). This suggests that plant-produced ethylene in shaded natural environments may not reach levels sufficient to induce a conspicuous response in neighboring r-plants. However, because ethylene production is dependent on biomass, large groups of shaded adult plants could produce localized pockets of high ethylene concentrations in natural settings, as reported [23,32]. In shaded environments dominated by shade-tolerant species, ethylene levels might be even higher, which could strongly influence the development of nearby small plants or seedlings. In high-density scenarios with high intense light (such as those found under proximity shade), the effect of plant-produced ethylene could be neutral or even positive, depending on the prevailing light intensity. However, if both light intensity and R/FR are reduced (canopy shade) [7], a negative impact on elongation is expected, suggesting that e-plants might use ethylene to regulate the growth of emerging seedlings, potentially reducing their elongation under these specific conditions to limit their growth and future competence for sunlight.

## 4. Materials and Methods

### 4.1. Plant Material and Growth Conditions

*Arabidopsis thaliana* Columbia (Col-0), *Cardamine hirsuta* Oxford (Ox), and tomato (*Solanum lycopersicum*) M82 and MicroTom accessions were used in this work, together with the *A. thaliana ein2-5* mutant and the *EBSn:GUS* marker line (kind gifts from the Alonso-Stepanova lab), that contain several copies of divergent DNA elements from the promoters of ethylene-inducible genes. Seeds were surface-sterilized with a 10% (*v*/*v*) sodium hypochlorite solution and 0.1% (*v*/*v*) of Tween 20 for 10 min, followed by 3–5 washes with sterile distilled water. Sterile seeds were sown on solid 0.5×MS- medium with vitamins and without sucrose. When indicated, the medium was supplemented with 10 μM ACC (1-aminocyclopropane-1-carboxylic acid). The media were plated on regular single-compartment Petri dishes, on two-compartment Petri dishes, or short wide-mouth sample glass bottles. After seed stratification at 4 °C for 3–5 days, plates were incubated in Aralab growth chambers at 22 °C, under continuous white light (W). For hypocotyl experiments, standard W had a Photosynthetic Photon Flux Density (PPFD) of around 50 μmol·m^−2^·s^−1^ and a R/FR of around 4. Other W intensities were achieved either by increasing the light intensity provided by the LED tubes or by dimming the standard W light covering the plates with layers of filter paper without altering light quality (i.e., same R/FR). Simulated shade (W+FR) was obtained by enriching W with supplementary FR light provided by LED lamps (www.philips.com), resulting in a similar PPFD but a lower R/FR (0.06) compared to our standard W conditions. Fluence rates were measured with a Spectrosense2 m associated with a 4-channel sensor (Skye Instruments Ltd. Llandrindod Wells, UK) which measures PAR (400–700 nm) and 10 nm windows in the blue (464–473 nm), R (664–673 nm), and FR (725–734 nm) regions [15].

### 4.2. Measurement of Hypocotyl Elongation

Seedlings were laid out flat on the plates, and the digital images were used to measure the length of hypocotyls with the National Institutes of Health ImageJ software (http://rsb.info.nih.gov/). At least 15 seedlings were used for each treatment with every genotype for every experiment. Statistical analysis of the data was performed using GraphPad Prism (www.graphpad.com/), selecting a suitable test for every set of data. Specifications of the tests used to analyze significances are given in the figure legends.

### 4.3. Ethylene Quantification

Leaves for ethylene quantification were obtained from plants grown in soil (10 × 10 cm square pots) at 22 °C under long-day conditions (8 h of darkness and 16 h of light, with a PPFD of around 135 μmol·m^−2^·s^−1^ and an R/FR of 4) for 4 weeks. Rosette leaves (in the case of *A. thaliana* and *C. hirsuta*) or leaflets of fully expanded tomato leaves were cut out from the plants, weighted, and placed inside 15 mL transparent glass vials that did not affect light quality. Closed vials containing the samples were placed horizontally under the same W or W+FR conditions for 24 h. In the case of seedlings, approximately 100 sterile *A. thaliana* seeds were sown on 0.5×MS- media in 50 mL short wide-mouth bottles. The bottles were closed with a transparent Petri dish lid and sealed with Micropore tape (3 M). Following stratification and incubation at 22 °C under W or W+FR conditions, the Petri dish lid was removed, and an airtight lid with a silicone septum was used instead to enable quantification of the ethylene produced by the seedlings in 24 h inside the bottle. The ethylene content in the vials (by detached leaves) and the bottles (by living seedlings) was measured using the ETD-300 ethylene detector (Sensor Sense, Nijmegen, The Netherlands). The data were normalized by sample weight and statistically analyzed with GraphPad Prism.

### 4.4. GUS Staining

EBSn:GUS seedlings, grown as mentioned in Figure 4, were transferred to a solution containing 90% (*v*/*v*) acetone and stored at 4 °C for at least 1 day. Seedlings then were washed twice with 50 mM Na_2_PO_4_ buffer at pH 7.2 at room temperature. Once washed, the seedlings were transferred to the GUS staining buffer (50 mM Na_2_PO_4_ buffer at pH 7.2, 1 mM EDTA, 0.1% (*v*/*v*) Triton X-100 and 0.5 mg·mL^−1^ of X-Gluc) and kept in darkness at 37 °C for 15 h. After this time period, the buffer was removed, and the seedlings were washed 3 times with solutions with different levels of ethanol (70%, 100%, and 50% (*v*/*v*), consecutively). Lastly, the seedlings were transferred to 50% (*v*/*v*) glycerol and placed into slides for visualization and image capturing.

## Figures and Tables

**Figure 1 plants-13-03212-f001:**
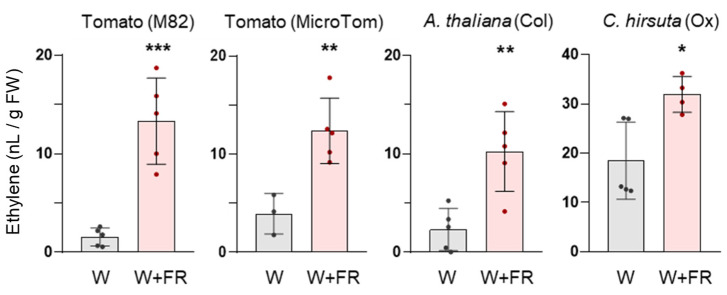
Simulated shade enhances ethylene production in different species. Detached leaves from the indicated plant species and accessions were placed in an airtight glass tube and exposed to either W or W+FR for 24 h. Bars show the mean amount of ethylene released to the headspace. Error bars correspond to the standard deviation (SD). Dots represent individual data points. Asterisks indicate significant differences (* *p* < 0.05; ** *p* < 0.01; *** *p* < 0.001) between the two light treatments, according to two-tailed Student’s t-tests for each species.

**Figure 2 plants-13-03212-f002:**
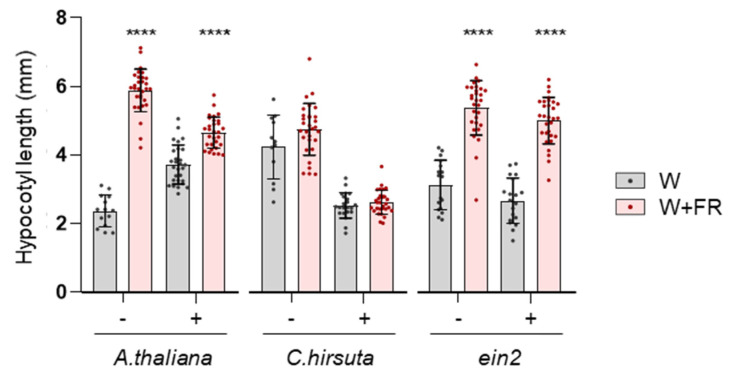
Enhanced ethylene production by ACC treatment alters hypocotyl elongation. Wild-type accessions of *Arabidopsis thaliana* (Col) and *Cardamine hirsuta* (Ox), as well as the ethylene-insensitive *A. thaliana* mutant *ein2-5*, were germinated and grown under W (50 μmol m^−2^·s^−1^) for 2 days in media supplemented (+) or not (−) with 10 μM ACC and then exposed for 5 days to either W or W+FR. The plots show mean and SD of hypocotyl length of approximately 20 seedlings per treatment. Dots represent individual data points. Asterisks indicate significant differences between the two light treatments, according to two-way ANOVA, followed by Sidak’s test (**** *p* < 0.0001). Non-significant differences are not indicated.

**Figure 3 plants-13-03212-f003:**
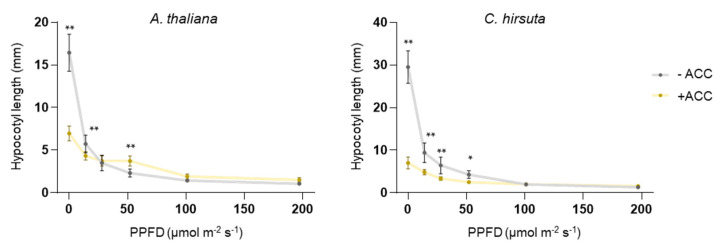
ACC supplementation impacts hypocotyl elongation in a light-dependent manner. *Arabidopsis thaliana* (Col) and *Cardamine hirsuta* (Ox) lines were germinated in media supplemented (+) or not (−) with 10 μM ACC and grown for 7 days under different W intensities ranging between 0 (darkness) and 200 μmol·m^−2^·s^−1^ of Photosynthetic Photon Flux Density (PPFD). The indicated W intensities used were the following (in order): 0, 14, 28, 52, 101, and 197 μmol·m^−2^·s^−1^. The plots show mean and SD of hypocotyl length of approximately 20 seedlings per treatment. Dots represent individual data points. Asterisks indicate significant differences (* *p* < 0.05; ** *p* < 0.01) between ACC-treated and untreated samples, according to two-way ANOVA, followed by Sidak’s test. Non-significant differences are not indicated.

**Figure 4 plants-13-03212-f004:**
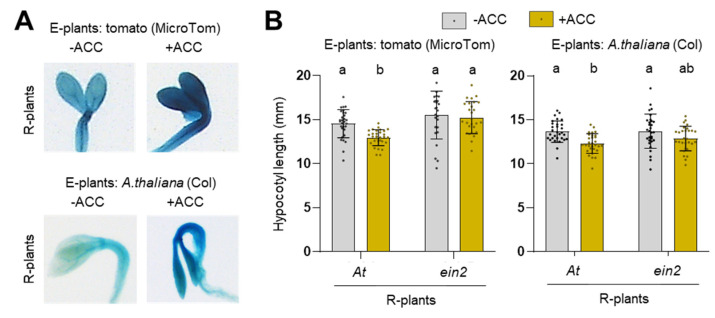
Ethylene produced by ACC-supplemented seedlings represses hypocotyl elongation of neighboring dark-grown seedlings growing without ACC. Plates with two separate compartments for media but a common airspace were used to grow e- and r-plants in different growth media. Tomato (*Solanum lycopersicum*) or *Arabidopsis thaliana* with (+) or without (−) ACC in one compartment as e-plants and different *A. thaliana* lines without ACC in the other compartment as r-plants. After stratification, plates were incubated in the dark for 4 days. (**A**) Representative GUS-stained seedlings of the ethylene reporter line *EBSn:GUS* grown in the r-plant compartment and exposed to volatiles from the indicated e-plants. (**B**) Barplots representing the mean and SD of hypocotyl length of at least 25 etiolated seedlings of *A. thaliana* WT and ethylene-insensitive *ein2-5* lines grown in the r-plant compartment and exposed to volatiles from the indicated e-plants. Dots represent individual data points. Different letters denote significant differences among means (two-way ANOVA, followed by Tukey test, *p*-value < 0.05).

**Figure 5 plants-13-03212-f005:**
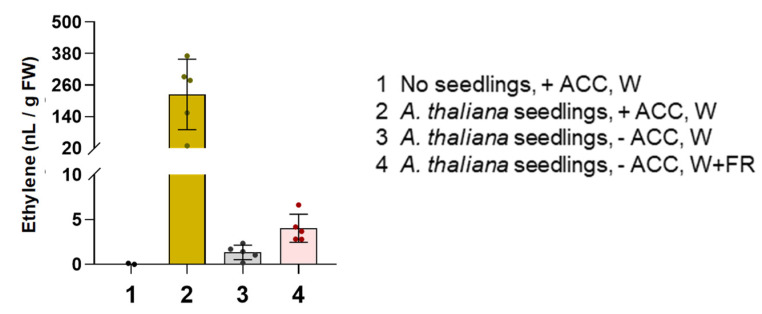
ACC supplementation causes a much stronger ethylene release than W+FR exposure. The plot shows the mean and SD of the amounts of ethylene released in 24 h by *Arabidopsis thaliana* Col-0 seedlings grown, as indicated, in glass bottles. Dots represent individual data points. Measurements carried out using the same containers, but lacking seedlings, are also shown as a control.

## Data Availability

The data and results presented in this study are available upon request from the corresponding authors.

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
