# Peer review of "Effect of Higher Ethylene Levels Emitted by Shade-Avoider Plants on Neighboring Seedlings"

_plants, 2024, doi:10.3390/plants13223212_

Round 1
Reviewer 1 Report
Comments and Suggestions for Authors
General Comments: The topic presented in the manuscript is actual and very interesting. The authors used some shade-avoider and shade-tolerant species to explore the roles of ethylene under different light conditions. Nevertheless, I think that the manuscript needs improvement before publishing. Please see the comments below.
Line 10-12 Please rephrase this sentence.
Many sentences may be long and complex, making them difficult to follow. Breaking these sentences down into shorter, more direct ones would make them easier to read (i.e., Lines 21-25; Lines 34-40; 65-68; 93-95; 133-137; 285-251).
Line 59 I recommend more examples of shade-tolerant species.
Line 64 phyB: Please write the full name.
Lines 65, 74 I recommend citing some references.
Lines 93-95 For other greenhouse plants whether this phenomenon is present, please give more examples.
Introduction: The research hypotheses have not been clearly formulated.
Lines 108-110 Please rephrase this sentence.
Lines 202-204 Please rephrase this sentence.
Figure 5 Please change the vertical label.
Please add more information on data analysis in the Materials and Methods section (i.e., software, variance analysis, significant analysis, duplication)
Overall, this study is an interesting subject. However, the manuscript should be thoroughly checked for the English language to improve sentence structure and flow.
Author Response
See attached PDF letter

Reviewer 2 Report
Comments and Suggestions for Authors
The problem of plant interactions is interesting and not well studied. The authors try to understand the role of the gaseous phytohormone ethylene in the interaction between shade tolerant plants and shade avoiding plants under low light conditions (at low R/FR values). Such conditions are usually created in shade trees. Shade tolerant plants emit consistently high amounts of ethylene under both white light and W+FR. Plants avoiding shade in this respect behave quite differently. The authors applied a variety of analytical methods. They used transgenic plants with the EBSn:GUS construct, ethylene-insensitive mutant, monitored the effect on hypocotyl growth in different setups, and continuously quantified ethylene release. An interesting experimental setup was devised to study the effect of ethylene released by plants on the growth of nearby plants.
The authors obtained compelling new evidence that ethylene may play a role as a signalling molecule in plant-plant communication between nearby plants.
In my opinion, the work is of high quality and may be of interest to plant biologists.
There are some minor comments to which I would like the authors to reply.
1. Does the quality of light change when passing through glass vials and bottles?
2. In experiments with cut leaves, ethylene release may be affected not only by light quality but also by the aging process of the separated leaves. 24 hours is a long time for cut leaves. The beginning of the senescence process can be checked e.g. by chlorophyll levels, but even better by the expression of genes activated by senescence - SAG genes.
3. EBSn:GUS it would be useful to indicate which gene promoter is cloned in this construct. Whole promoter or specific cis-elements introduced into transgenic plants?
4. Lines 120-121. It seems that the authors did not express their idea correctly. Judging from the results of Fig. 1, shade tolerant plants secrete more ethylene at W+FR than shade avoiding plants. Their ethylene formation was not impaired, but they responded more weakly to W+FR light.
6. In Fig. 3, for points 2 and 3, hypocotyl lengths are not plotted on the abscissa axis with exact PPFD values. This is the most dynamic zone and accurate PPFD values would be helpful.
Author Response
See attached PDF letter

Round 2
Reviewer 1 Report
Comments and Suggestions for Authors
The authors have mostly addressed my comments and suggestions for improvement and/or correction and significantly improved the manuscript compared to the original version. However, I have some comment as follows.
Line 110 I recommend replacing “To address this question” with “To explore potential mechanism”
Fig.1; Fig. 5 Please write the label (nL ethylene….) properly.
Author Response
- We have highlighted the changes incorporated to the manuscript, as indicated in the received e-mail.
R1.1. Line 110 I recommend replacing “To address this question” with “To explore potential mechanism”
- Answer. We have modified the sentence, as suggested.
R1.2. Fig.1; Fig. 5 Please write the label (nL ethylene….) properly.
- Answer. We have unified how we present the labels of the y-axis in both figures [Ethylene (nL / g FW)] – thanks for noticing.
- We have also updated the acknowledgement section.